# Effect of Intermediate Airway Management on Ventilation Parameters in Simulated Pediatric Out-of-Hospital Cardiac Arrest: Protocol for a Multicenter, Randomized, Crossover Trial

**DOI:** 10.3390/children10010148

**Published:** 2023-01-12

**Authors:** Loric Stuby, Elisa Mühlemann, Laurent Jampen, David Thurre, Johan N. Siebert, Laurent Suppan

**Affiliations:** 1Genève TEAM Ambulances, Emergency Medical Services, CH-1201 Geneva, Switzerland; 2ESAMB-École Supérieure de Soins Ambulanciers, College of Higher Education in Ambulance Care, CH-1231 Conches, Switzerland; 3Ambulances de la Ville de Sion, Emergency Medical Services, CH-1950 Sion, Switzerland; 4Division of Pediatric Emergency Medicine, Geneva Children’s Hospital, Geneva University Hospitals, CH-1205 Geneva, Switzerland; 5Division of Emergency Medicine, Department of Anesthesiology, Clinical Pharmacology, Intensive Care and Emergency Medicine, Geneva University Hospitals, CH-1211 Geneva, Switzerland

**Keywords:** emergency medical services, airway management, laryngeal masks, i-gel, out-of-hospital cardiac arrest, cardiopulmonary resuscitation, non-invasive ventilation, pediatrics

## Abstract

Most pediatric out-of-hospital cardiac arrests (OHCAs) are caused by hypoxia, which is generally consecutive to respiratory failure. To restore oxygenation, prehospital providers usually first use basic airway management techniques, i.e., bag-valve-mask (BVM) devices. These devices present several drawbacks, most of which could be avoided using supraglottic airway devices. These intermediate airway management (IAM) devices also present significant advantages over tracheal intubation: they are associated with higher success and lower complication rates in the prehospital setting. There are, however, few data regarding the effect of early IAM in pediatric OHCA. This paper details the protocol of a trial designed to evaluate the impact of this airway management strategy on ventilation parameters through a simulated, multicenter, randomized, crossover trial. The hypothesis underlying this study protocol is that early IAM without prior BVM ventilations could improve the ventilation parameters in comparison with the standard approach, which consists in BVM ventilations only.

## 1. Introduction

### 1.1. Background

Survival rates after pediatric out-of-hospital cardiac arrest (OHCA) are still excessively low despite many advances in resuscitation science over the past decades, with recent studies reporting rates of survival to hospital discharge ranging from 0.0% to 21.2% [1,2,3]. Generally, pediatric OHCA is secondary to hypoxia [4], with sudden infant death syndrome, trauma and drowning being the most common etiologies [5]. Restoring oxygenation is, therefore, of paramount importance to achieve the return of spontaneous circulation (ROSC). Consistently, delays in intermediate or advanced airway management are associated with lower survival rates [6]. There are, however, few data regarding the effects of specific airway management strategies in the case of pediatric OHCA [7,8].

Generally, emergency medical services first use basic airway management techniques, i.e., bag-valve-mask (BVM) devices, to restore oxygenation after pediatric OHCA [9]. These devices present several drawbacks. Difficulties in ventilating with a BVM are not uncommon [10], with a significant incidence of excessive or insufficient (in terms of volume and/or pressure) ventilations [11]. Air leaks are frequently encountered when using basic airway management devices and are associated with lower and often unsatisfactory tidal volumes [12]. Gastric insufflation is also common when such devices are used [13]. The gas insufflated in the stomach can alter oxygenation by restricting the total lung capacity and, consequently, lung compliance [14]. Since decreased lung compliance requires the use of higher pressures to obtain similar tidal volumes, gastric inflation can indirectly impair venous return [14,15,16]. Moreover, chest compressions must be interrupted to provide ventilations when basic airway management devices are used. These interruptions decrease coronary and cerebral blood flow and should be minimized, as they have been associated with lower survival rates [17].

In contrast to basic airway management, advanced airway management, i.e., tracheal intubation, provides optimal airtightness—thereby avoiding gastric inflation and limiting the risks associated with regurgitation—while allowing the provision of asynchronous ventilations during cardiopulmonary resuscitation (CPR). However, tracheal intubation requires advanced skills that must be maintained through regular practice [18,19]. Depending on the regional context, skilled prehospital providers may not be immediately available, if at all. This is particularly important when taking care of critically ill children, whom many consider difficult to intubate [20,21]. The failure rate of tracheal intubation at first attempt is high in pediatric cardiac arrest, even in the hospital setting [22], and multiple tracheal intubation attempts are associated with worse neurological and survival outcomes [23]. In addition, tracheal intubation takes time to prepare, and there is usually a delay before the procedure can be performed.

Since both basic and advanced airway management devices present significant limitations, intermediate airway management (IAM), i.e., the use of supraglottic airway devices [24], could represent a valuable alternative. One of the most studied IAM devices is i-gel^®^, which is both easy and fast to insert [25,26]. This device also provides high leak pressures [27] and is associated with high overall success rates [27,28,29]. Its insertion procedure is simple and easy to remember [30]. Regurgitation and aspiration are not more frequent with IAM devices than with tracheal intubation [31] and are much less likely than when basic airway management techniques are used [32]. In most cases, the use of i-gel^®^ enables continuous chest compressions [33] and a higher rate of successful initial ventilation [31]. Early i-gel^®^ insertion also increased the chest compression fraction (CCF) and improved the ventilation parameters in an adult model of OHCA [34,35,36]. There was no difference in neurological outcomes after OHCA in adults whose airways had been managed using IAM devices in comparison with those who had undergone tracheal intubation [31,37].

There is increasing evidence that IAM devices can be safely used in children. In two pediatric OHCA studies, American paramedics had significantly higher success rates with IAM devices than with tracheal intubation [38,39]. A neonatal animal model showed that the use of such devices while providing continuous chest compressions was feasible and non-inferior to tracheal intubation [40]. Recently, a registry-based study reported poorer outcomes when emergency physicians chose to attempt tracheal intubation rather than using IAM devices [41]. There are, however, few data regarding the impact of IAM on ventilation parameters during pediatric OHCA. The hypothesis underlying this study is that, in the case of pediatric OHCA, compared with the standard approach consisting in BVM ventilations only, the early insertion of an i-gel^®^ device with no prior BVM ventilation should enhance the ventilation parameters.

### 1.2. Objectives

The primary objective of this study will be to assess whether in a simulated manikin-based pediatric OHCA model, the immediate insertion of i-gel^®^ followed by asynchronous ventilations (experimental approach) improves the ventilation parameters compared with the standard approach according to the guidelines of the American Heart Association (AHA) [9].

The secondary objective will be to examine the impact of these approaches on the CCF and on chest compression quality (rate, depth, and chest recoil).

## 2. Materials and Methods

### 2.1. Study Design and Setting

We will carry out a prospective, superiority, multicenter, simulation-based, randomized, crossover trial. This study will take place in Swiss emergency medical services. Figure 1 shows the study flowchart, and Figure 2 details the study sequence. The trial schedule is displayed in Table 1. The study will be conducted in accordance with the extension to randomized crossover trials of the Consolidated Standards of Reporting Trials (CONSORT) guidelines [42] and will comply with the Reporting Guidelines for Health Care Simulation Research [43]. The present study protocol adheres to the Standard Protocol Items: Recommendations for Interventional Trials (SPIRIT) 2013 Checklist (Appendix A) [44]. The regional ethics committee confirmed that this study does not fall within the scope of the Swiss legislation regulating research on human subjects (CCER-Req-2022-00859). The trial will be carried out according to the principles of the Declaration of Helsinki [45] and to the Good Clinical Practice guidelines [46]. 

The Swiss prehospital setting has been extensively described in prior publications [34]. Briefly, the Swiss system varies considerably from one region to another. Prehospital emergency physicians are not always available, if at all, in certain regions. Swiss paramedics are, however, almost always available to respond to life-threatening emergencies. These advanced paramedics graduate from Colleges of Higher Education in Ambulance Care after a 3-year (5400 h) training curriculum covering most aspects of prehospital emergency care. Emergency Medical Technicians (EMTs) are granted a federal degree after a single year of training (1800 h). Typically, the first professional rescuers dispatched to take care of OHCA victims are either two paramedics or a paramedic and an EMT.

### 2.2. Participants, and Inclusion and Exclusion Criteria

Registered EMTs and paramedics actively working in any of the participating trial centers will be eligible for inclusion. All these centers use i-gel^®^ devices in their clinical practice. EMTs will be randomly excluded if there are more EMTs than paramedics. There will be no other exclusion criteria. Participants will be recruited with the help of a local study coordinator using a standardized email template (Appendix A) which will provide the participants with all necessary information regarding the study, including data reuse and protection policy. Participants will be informed that the study is about pediatric OHCA management but will be blinded as to the study outcomes to prevent preparation bias. Each participant will be free to participate and shall withdraw at any time without giving a reason. The benefit to participants will be an increase in their knowledge of the i-gel^®^ device, which may prove useful in their daily practice. There will be no financial incentive. The invitation email will contain a link to the study platform. This platform will host an online survey form designed to gather demographic data and obtain the participants’ consent. Participants will be provided with a generic email address directed both to the main investigator and the investigator who created the platform, to enable them to ask further questions. 

All participants will be assumed to have similar skills in BVM ventilation and i-gel^®^ use, as this is part of their regular practice and training.

### 2.3. Intra-Cluster Randomization

Two levels of randomization will be used. Each trial center will represent a cluster, and teams will first be randomized intra-cluster. This will be carried out using an online balanced team generator [47], with stratification by professional status. There will be at least one paramedic per team to be consistent with current clinical practice. In mixed teams (i.e., paramedic and EMT), the “team leader” will always be the paramedic. In paramedic-only teams, the role of each paramedic will be left to their discretion as in actual clinical practice. Team leaders will maintain their role throughout all scenarios. 

### 2.4. Self-Managed Training

The participants will have 30 min of self-managed training. This training will be supported by a standardized demonstration video of the experimental approach. They will have full control over the video and will be able to watch it at will during this 30 min training period.

### 2.5. Study Groups, Second Randomization and Concealment of Allocation

The second level of randomization will take place immediately after the self-managed training session. The teams will be randomized to one of the two study paths by opening an opaque, sealed envelope conceived using a block randomization list generated online by Loric Stuby (L.St.) using a 1:1 ratio, with block sizes of 2 and 4 and stratification according to the trial centers [48]. This stratification will be used to account for differences in the approaches of the participants (e.g., initial airway management strategy, task distribution, local procedures during CPR, quality processes or prior specific training). 

The AHA guidelines will be considered as standard approach [49]. These guidelines recommend alternating 15 compressions and 2 ventilations, starting with compressions [9].

The experimental approach consists in the immediate insertion of an i-gel^®^ device with no prior ventilation. Chest compressions are delivered continuously once the cardiac arrest has been identified. Once i-gel^®^ has been inserted, ventilations should be performed at a rate of 20 to 30 per minute, as recommended by the AHA [9].

### 2.6. Manikin and Resuscitation Equipment

The same high-fidelity Wi-Fi manikin and dedicated multiparametric monitor/defibrillator (Laerdal SimBaby; Laerdal Medical, Stavanger, Norway) will be used throughout the study. The manikin will automatically record all study outcomes (e.g., chest compression rate, depth, recoil, ventilation volume and rate, etc.) except for the time point of epinephrine injection, which will be manually tagged. SimBaby is a realistic manikin representing a 9-month-old child that has a height of 71 cm. The manikin weighs 4.9 kg (actual weight). To be consistent with the age, the simulated child weight will be told to be 9 kg according to the appropriate Best Guess formula (0.5 × age in months + 4.5) [50]. Teams will have access to their usual resuscitation equipment (e.g., they may choose to use a nasopharyngeal cannula). The decision to use a specific item will remain at the discretion of the teams, as in a real resuscitation. A back compensation, using a folded blanket, will already be in place. An i-gel^®^ device size 1.5 (Intersurgical Ltd., Wokingham, UK) and a tube of lubricant recommended by the manikin manufacturer will be available in the intervention bag. Only the correct BVM (both bag and mask) size will be available. An overview of the manikin’s characteristics and of the features of the multiparametric monitor will be provided when entering the study room through a standardized video. 

### 2.7. Pediatric Cardiac Arrest Scenario

The team will perform two consecutive 10 min, realistic pediatric OHCA scenarios. To ensure that each participant is exposed to the same scenario, with similar challenges in decision making and treatment provided, the procedure will be standardized on all sites. This should also help to minimize confounders by delivering uniform and consistent information. The first therapeutic action (i.e., ventilation or chest compression) will be defined as T0. Participants will be informed that each resuscitation scenario will be discontinued after 10 min, regardless of their decisions, and that no feedback will be given. To increase the level of fidelity, two stressors will be used. Firstly, a simulated parent, who will be played by the same investigator (E.M.) during all scenarios, will request information by asking “What is going on?” 3 times (at T0 + 2 min, T0 + 4 min and T0 + 8 min). Secondly, traffic noises [51] will be played during the scenario. The simulated parent will be able to communicate the child’s weight (9 kg), age (9 months) and height (71 cm), as well as the fact that the child has no known health problems and was born at term by vaginal route with a normal birth weight and that there were no post-partum complications.

The scenario will start with a clinical description to expose the life-threatening condition of the patient. One of the investigators will state: *“Here is Cayla, a 9-month-old child who was left unattended for 10 min. She was found cyanotic by her parent who called the emergency call center. A few minutes later, she collapsed. She is now unconscious, limp, cyanotic, and apneic. A medical mobile unit has already been dispatched and will be on site in about ten minutes.”* A rephrasing will be asked of the team leader to ensure correct understanding (closed-loop communication). The team leader will then pick up the top envelope of the stack, and open it to discover the path to follow, starting either with the standard or experimental approach. The simulated child will be apneic and pulseless if checked. After placing the pads, asystole will be displayed on the defibrillator’s screen. CPR waves will be automatically shown when compressions are delivered to increase the fidelity of the simulation. Regardless of the treatment given, all subsequent rhythm analyses will show refractory asystole. Participants will be able to successfully obtain intraosseous/intravenous access on their first attempt. There will be no further intervention or educational follow-up after the study period. Team members will be asked to alternate their position after every 2 min cycle. At the end of the scenario (T0 + 10 min), participants will be shown a normal sinus rhythm, and ROSC will be present. All the equipment will be fully restored. Then, the same scenario will be played a second time using the other airway management strategy, with no more interactions between the study team and the participants nor between the participants. 

### 2.8. Outcomes

The primary outcome will be alveolar ventilation. It will be determined by subtracting the dead space volume from each ventilation. The simulated child’s dead space volume corresponds to about 27 mL using the formula proposed by Numa and Newth [52]. The physiological tidal volume range is 5 to 8 mL/kg [53], corresponding to 45 to 72 mL for the simulated child.

The secondary outcomes will be:-The proportion and the number of ventilations below (<45 mL), within (45–72 mL) and over (>72 mL) the target volume;-The time to the first efficient ventilation (≥45 mL);-The time to the first compression;-The CCF;-The chest compression rate;-The proportion of chest compressions below (<100/min), within (100–120/min) and over (>120/min) the target rate [54];-The chest compression depth;-The proportion of chest compressions below (<4.3 cm) and within (≥4.3 cm) the target depth [54]; this threshold corresponds to one-third of the manikin’s measured anteroposterior chest depth;-The proportion of chest compressions below (<3 cm) and within (>3 cm) the manufacturer’s target;-The proportion of complete chest recoil;-The time to the first epinephrine injection;-The proportion of scenarios in which epinephrine is administered within 5 min [9].

### 2.9. Sample Size Calculation

The sample size is based on an estimate using data from two previous simulation studies [35,49]. Ventilation is expected to be provided for 8 min only (given the time necessary to prepare ventilation devices). In an adult scenario lasting 10 min, there were 39 ventilations in the IAM group (with a target rate of 10/minute) and 19 in the BVM group [35]. In this study, targeting a ventilation rate of 20–30/minute, around 100 ventilations are expected in the IAM group. In the BVM group, 40 ventilations are to be expected if a ratio of 15 compressions to 2 ventilations is applied. The tidal volume should be similar in both groups, around 52 mL worth 25 mL of alveolar ventilation. The expected difference in alveolar minute ventilation between basic airway management and IAM devices is 185 mL (125 mL versus 310 mL). The standard deviation is estimated to be 140 mL [49]. To be conservative, the correlation used to compute the calculation is 0. With an alpha set at 5% and a power of 90%, 15 teams (30 simulations) are required. Given the harmless study design (simulation), more groups will be accepted if available.

### 2.10. Blinding, Bias Minimization, and Data Collection and Extraction

To prevent assessment bias, data will be gathered automatically through the manikin’s sensors, and extracted to a comma-separated values (CSV) file. Variables of interest will be generated using a custom-coded PHP script. Demographics and consent will be collected using an electronic form hosted on a Joomla 4.2 website (Open-Source Matters, New York, NY, USA). Missing data will be treated as such. No imputation technique will be used. The curated dataset will be sent in DTA file format to the data analyst, who will be blinded as to group allocation (all data that could potentially lift the blinding will be removed). The “.dta” file format is a proprietary binary format designed to be used with Stata data analysis software. All investigators will have full access to the curated data file. 

### 2.11. Statistical Analysis

Data distribution will be assessed graphically, and when in doubt, using the Shapiro-Wilk test. The variables will be described accordingly using either means (standard deviations and/or 95% confidence intervals) or medians (quartiles). 

The dependency of the variables, due to the crossover design, will be taken into account either using paired tests (paired Student’s *t*-test, Wilcoxon matched-pairs signed-rank test or sign test depending on the assumptions) or by applying non-paired tests on the difference (Student’s *t*-test, or Wilcoxon–Mann–Whitney test). Analyses will be performed on an intention-to-treat basis.

A two-sided *p*-value lower than 0.05 will be considered significant. The data will be analyzed using Stata V15.1 (StataCorp-LLC, College Station, TX, USA).

### 2.12. Confidentiality

Study participants’ information will be kept confidential. Data regarding study participants will be coded. No identifying information will be revealed or transferred to third parties.

### 2.13. Criteria for Discontinuing or Modifying Interventions

Each team will be allocated two 30 min slots. The first slot will consist in the self-managed training session. The second slot should enable participants to complete the two 10 min scenarios.

There will be no interim analysis since the study will take place over a limited period and because no harm could result from simulation-based CPR scenarios.

## 3. Results

### 3.1. Milestones

The findings will be presented at scientific congresses and submitted for publication in a peer-reviewed, international journal, regardless of the results. It should be submitted in the first half of 2023. 

### 3.2. Protocol Version

The current version of this protocol is 1.0 (5 December 2022). Study sessions are expected to take place in January and February 2023. All protocol modifications will be clearly detailed in the final manuscript.

### 3.3. Trial Registration

This study is registered at https://clinicaltrials.gov/ (accessed on 10 August 2022) under the trial registration number NCT05498402.

### 3.4. Data Curation and Availability

The curated database will be deposited on Yareta [55] and will be immediately available before the final manuscript will be submitted for peer review. The original documents and files will be safely stored.

## 4. Discussion

The possibilities of addressing some of the knowledge gaps identified by the International Liaison Committee on Resuscitation (ILCOR) through clinical studies are particularly limited when the pediatric population is involved [56]. In this context, simulation-based studies could provide significant data. The protocol outlined in this manuscript could provide useful evidence and help to determine the most appropriate airway management strategy.

The main limitation of this study is that it will be conducted on a manikin. Therefore, there will be no direct evidence regarding actual clinical outcomes. Nevertheless, should different airway management strategies lead to different ventilation parameters, the same should be true in real clinical situations. However, the successful insertion rates could differ between simulated and actual OHCAs. 

In this trial, pediatric OHCA scenarios will be managed by two providers only. Actual clinical outcomes might prove different depending on the presence of BLS-trained bystanders, of prehospital physicians, or of other prehospital providers. The presence of trained rescuers can indeed enable the team leader to assign tasks, such as high-quality chest compressions, to these additional providers, thus allowing the ambulance crew to concentrate on other aspects of CPR management (such as airway management, crisis resource management, parent inclusion, rhythm analyses and ALS treatments). This could impact CPR quality because of fatigue, organizational aspects and differences in cognitive load, thereby limiting the generalization of the results gathered during this study. 

Another limitation is that the manikin could prove to be more difficult to ventilate with i-gel^®^ devices than actual patients. Indeed, airtightness improves over time when i-gel^®^ devices are used [57]. This is due to the thermoplastic properties of the non-inflatable cuff, which forms a progressively more efficient seal around the larynx after warming to body temperature. A field trial could, therefore, be considered according to the results that will be gathered during this study. 

While technical issues such as computer failure or manikin breakdown could prevent us from gathering the complete data (unavoidable attrition), quality control checks will be performed prior to each study session to limit the occurrence of such issues.

The strengths of this protocol are the use of the SPIRIT guidelines, the randomized crossover design, the specific and little-explored outcomes studied and the multicenter design. By comparing two airway management strategies, this study could help to address an evidence gap regarding the optimal airway management strategy in the case of pediatric OHCA. Its results could support the external validity and implementation of the most efficient airway management strategy in current practice.

## 5. Conclusions

This study could help to ascertain whether early IAM using an i-gel^®^ device improves the ventilation parameters in a pediatric simulated model of OHCA. The results generated by this trial could help to determine the most appropriate airway management strategy in this uncommon, yet critical situation.

## Figures and Tables

**Figure 1 children-10-00148-f001:**
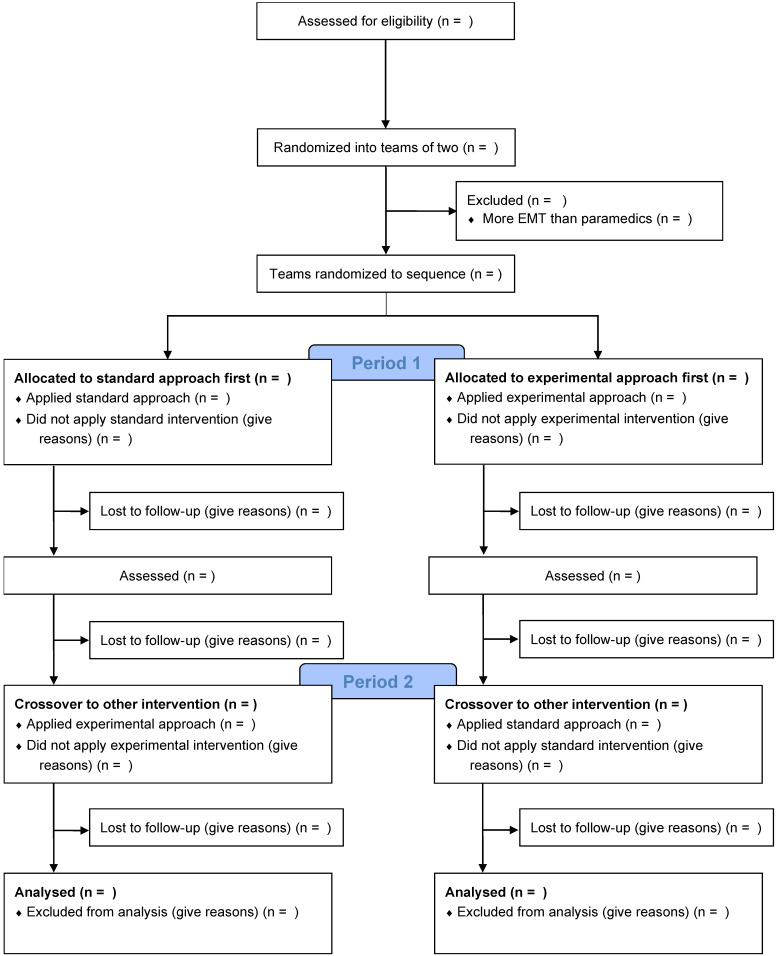
Study flowchart.

**Figure 2 children-10-00148-f002:**
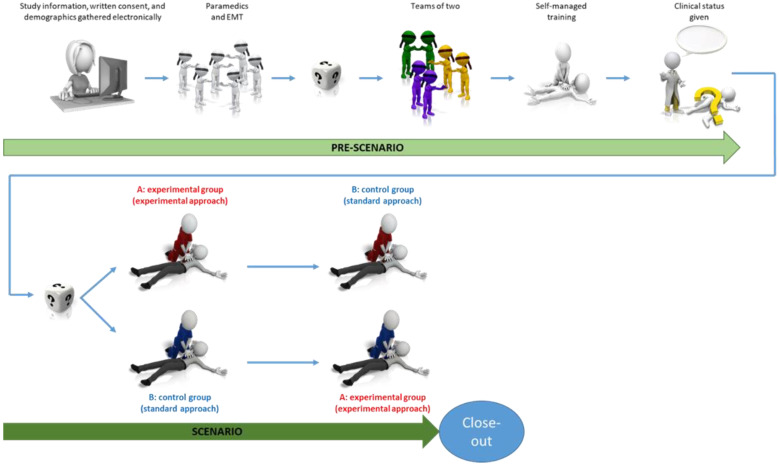
Study sequence.

**Table 1 children-10-00148-t001:** Trial schedule.

	Study Period			
	Enrolment	Intervention		Close-Out
Timepoint	*t* _0_	*t* _1_	*t* _2_	
**STUDY PROCEDURES**				
Recruitment and consent	✓			
Eligibility check	✓			
Teams of two	✓			
30 min self-managed training	✓			
**ASSESSMENTS**				
Participants’ characteristics	✓			
Randomization	✓			
Standard approach		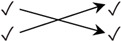		✓
Experimental approach				✓
✓: Performed				

## Data Availability

Not applicable.

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
