# Peer review of "Effect of Intermediate Airway Management on Ventilation Parameters in Simulated Pediatric Out-of-Hospital Cardiac Arrest: Protocol for a Multicenter, Randomized, Crossover Trial"

_children, 2023, doi:10.3390/children10010148_

Round 1

Reviewer 1 Report

Dear Authors,

Excellent scientific work. My decision is accept in present form.

Reviewer 2 Report

This is a clinically important, perfectly designed study.

No doubt, I highly recommend  its publication  waiting for its results.

Congratulations for all study contributors!

One remark- line 241 CCF abbrev should be explained.

Reviewer 3 Report

Dear Editor,

Many thanks for asking me to review this very interesting paper. The authors have planned this research very well and I look forward to seeing the final results of the study. I have made immense effort to find mistakes in this paper, but I can’t seem to find any. It is very well written and I have enjoyed reading this paper. It seems logical that I-gel is a definitive airway as it eliminates jaw thrust and two-person ventilation, but research to prove this or disprove this will be useful. It will be useful to know the thoughts of participants preference, BMV or i-gel. I am not sure if this is within the scope if the study My personal preference will be i-gel.

I have few suggestions; I hope the authors may find this useful

 -          can you reduce the number of abbreviations

-          Line 94, Line 102- please mention the simulation will be done using a manequin

-          Line 119, 121- what does 5400h, 1800h mean; is it hours? If so, please mention hours

-          Section methods, Figure 1, section 2.2: flow chart: what are the exclusion criteria?

-          Section methods, Figure 1: flow chart: please mention in the foot note, standard method = BMV, experimental approach =LMA, if this is what authors imply

-          Line 275: expand DTA, should this be .dta ?

-          The Stata_dta format (with extension. dta) is a proprietary binary format designed for use as the native format for datasets with Stata, a system for statistics and data analysis

-          Line 281: for the benefit of reader of every level please expand SD – standard deviation, CI – Confidence interval; Q1, Q3 - quartiles

-          Line 339: airtightness, should this be air tightness?

Author Response

Point 1: Dear Editor,
Many thanks for asking me to review this very interesting paper. The authors have planned this research very well and I look forward to seeing the final results of the study. I have made immense effort to find mistakes in this paper, but I can’t seem to find any. It is very well written and I have enjoyed reading this paper. It seems logical that I-gel is a definitive airway as it eliminates jaw thrust and two-person ventilation, but research to prove this or disprove this will be useful. It will be useful to know the thoughts of participants preference, BMV or i-gel. I am not sure if this is within the scope if the study My personal preference will be i-gel.

I have few suggestions; I hope the authors may find this useful
Response 1: Thank you very much for your constructive feedback and for the time and effort you spent reviewing our protocol. Regarding participants’ preference, we assessed the providers’ satisfaction in a prior study (https://www.mdpi.com/2077-0383/11/1/217), and found that it was higher in the i-gel group (p=0.01).
Point 2: can you reduce the number of abbreviations
Response 2: Thank you for this comment. We have made some adaptations to reduce the number of abbreviations. EMS  Emergency Medical Services, BAM  basic airway management, and TI  tracheal intubation are now given in their unabbreviated form. The abbreviations OHCA, ROSC, BVM, CPR, CCF, BLS, ALS, ILCOR, and AHA are commonly used in resuscitation science and among the health care community, so we have decided to leave them in their current, abbreviated form. CONSORT and SPIRIT are also commonly used abbreviations, and PHP is most used in its contracted form; therefore, these abbreviations were also kept. We also decided to keep EMT in its abbreviated form because this abbreviation was used in the figures. Nevertheless, SD and CI have been expanded in line with your comment
(see response 9).
The only non-common abbreviation we kept was IAM for intermediate airway management, because it is a new concept that we wanted to spread (https://www.mdpi.com/2227-
9032/10/5/961).
Point 3: Line 94, Line 102- please mention the simulation will be done using a manequin
Response 3: We have added this clarification. The sentence is now:
“The primary objective of this study will be to determine whether, in a simulated manikinbased pediatric OHCA model, […].”
Point 4: Line 119, 121- what does 5400h, 1800h mean; is it hours? If so, please mention hours.
Response 4: Yes, those were the hours we wanted to mention. In line with your comment, the
sentence has been clarified thus: 

“These advanced paramedics graduate from Colleges of Higher Education in Ambulance Care after a 3-year (5400 hours) training curriculum covering most aspects of prehospital emergency care. Emergency Medical Technicians (EMTs) are granted a federal degree after a single year of training (1800 hours).”
Point 5: Section methods, Figure 1, section 2.2: flow chart: what are the exclusion criteria?
Response 5: Thank you for spotting this inconsistency between the Figure and the text. We have corrected the text to match the Figure: EMTs will be randomly excluded if there are more EMTs than paramedics. There will be no other exclusion criteria.”
Point 6: Section methods, Figure 1: flow chart: please mention in the foot note, standard method = BMV, experimental approach =LMA, if this is what authors imply
Response 6: This is not exactly what we have implied. In the experimental approach, providers will be asked to apply continuous chest compressions since the beginning of the resuscitation sequence. Therefore, the approaches cannot be summarized solely by the airway management strategy used. Therefore, we would rather refrain from adding this simplification to avoid confusion.
Point 7: Line 275: expand DTA, should this be .dta ?
Response 7: We added a clarification (please see response 8 below).
Point 8: The Stata_dta format (with extension. dta) is a proprietary binary format designed for use as the native format for datasets with Stata, a system for statistics and data analysis.
Response 8: A clarification was made regarding the DTA format. The paragraph now reads:
“The curated dataset will be sent in DTA file format to the data analyst, who will be blinded as to group allocation (all data that could potentially lift the blinding will be removed). The “.dta” file format is a proprietary binary format designed to be used with the Stata data analysis software. All investigators will have full access to the curated data file.”
Point 9: Line 281: for the benefit of reader of every level please expand SD – standard deviation, CI – Confidence interval; Q1, Q3 - quartiles
Response 9: Even though SD, and 95% CI are commonly used in the scientific community, and the targeted audience of the journal, we agree to add the expansion. The sentence is now:
“The variables will be described accordingly using either mean (standard deviation and/or 95% confidence interval) or median [quartiles].”
Point 10: Line 339: airtightness, should this be air tightness?
Response 10: We believe that it is correctly spelt in the current version of the manuscript (https://www.merriam-webster.com/dictionary/airtight). The words are used twice, current lines 59 and 346 (ex 339) in one word. 
